# Other Language Proficiency Predicts Unique Variance in Verbal Fluency Not Accounted for Directly by Target Language Proficiency: Cross-Language Interference?

**DOI:** 10.3390/brainsci9080175

**Published:** 2019-07-24

**Authors:** Kenneth R. Paap, Lauren A. Mason, Brandon M. Zimiga, Yocelyne Ayala-Silva, Matthew M. Frost, Melissa Gonzalez, Lesley Primero

**Affiliations:** Department of Psychology, San Francisco State University, 1600 Holloway Avenue, San Francisco, CA 94132, USA

**Keywords:** bilingualism, verbal fluency, cross-language interference, inhibition

## Abstract

The purpose of the study was to investigate cross-language effects in verbal fluency tasks where participants name in English as many exemplars of a target as they can in one minute. A series of multiple regression models were used that employed predictors such as self-rated proficiency in English, self-rated proficiency in a language other than English, a picture naming task used to measure productive vocabulary, the percentage of English use, and the frequency of language switching. The main findings showed that self-rated proficiency in the non-English language accounted for unique variance in verbal fluency that was not accounted for directly by self-rated proficiency in English. This outcome is consistent with cross-language interference, but is also consistent with an account that assumes bilingual disadvantages in verbal fluency and picture naming are due to bilinguals having weaker links between semantic concepts and their phonological form. The present study is also discussed in terms of a broader framework that questions whether domain-general inhibition exists and also whether it plays an important role in bilingual language control.

## 1. Introduction

The purpose of this Special Issue is to investigate “the influences of the native language on the second language and, reciprocally, of the second language on the native language in bilinguals.” This motivated us to revisit a dataset that formed the basis for a large-scale investigation of the bilingual advantage in switching cost hypothesis [1]. In our earlier study, there were no bilingual advantages in switching costs or mixing costs across three different cued switching tasks. However, for present purposes, the important data derives from a set of verbal fluency tasks that were also completed by all participants. In a verbal fluency task, participants were asked to generate as many instances of a category as they could in one minute. Overall, there was a statistically significant bilingual disadvantage that was relatively small as an effect size, F(1,203) = 6.06, partial η^2^ = 0.029. Both category fluency (e.g., word targets like musical instruments or vegetables) and letter fluency (e.g., words beginning with F or A) were tested in English. Like the vast majority of studies (see [1], Table 2) testing for this interaction, there was no Task (category versus letter) × Group (monolingual versus bilingual) interaction.

Why do bilingual disadvantages in verbal fluency occur? The explanation that seems to have the most support is cross-language interference between the two languages [2]. That is, according to the “interference” hypothesis bilinguals may fail to access some instances, even though they are available (i.e., have been acquired), because competition from the nontarget lexicon interferes with their retrieval (we are referring here to the classic distinction made by Tulving [3] between memory items that are available, but not necessarily accessible in a specific context). However, competing hypotheses are also in play. If bilinguals have smaller vocabularies in the test language, then this could contribute to the bilingual disadvantage. Similarly, if the links between semantic concepts and their corresponding articulatory phonology are weaker in bilinguals (because those links have been traversed less often), then this could also contribute to the bilingual disadvantage in verbal fluency.

How might one develop the interference account of the bilingual disadvantage in verbal fluency? Logically, a strong predictor of verbal fluency in English would be proficiency in English, especially lexical knowledge. Participants with greater lexical knowledge in English should generate more correct responses in the verbal fluency tasks. However, if there is cross language interference, then the amount of interference from an L2 on L1 should increase as L2 proficiency increases. This follows from the common-sense notion that practice and resulting gains in proficiency lead to stronger competition from L2. For a native English speaker learning Spanish as an L2, the competition to /kæt/ from the phonological form /gato/ increases as its direct connection to the semantic concept CAT grows stronger. These simple assumptions predict that, when English is the test language, the strength of the link between semantics and phonological form in the Other (For conciseness we will refer to the non-English language of a bilingual as the Other language and their degree of proficiency as Other Proficiency.) language should predict unique variance in verbal fluency performance apart from that accounted for by the degree of proficiency in English.

The weaker links account of bilingual disadvantages in picture naming and verbal fluency would make the same prediction, namely, that L2 proficiency has a negative effect on L1 verbal fluency and accounts for unique variance beyond that accounted for directly by L1 proficiency. The weaker links account relies on the cumulative history of language use prior to a verbal fluency test. That is, using L2 steals time from using L1 and results in L1 links that are weaker in bilinguals compared to monolinguals. Furthermore, as L2 use increases (and L2 proficiency improves) the strength of the L1 links will lag further behind those for monolinguals. Thus, the weaker links perspective also predicts that increases in L2 proficiency will be associated with decreases in L1 verbal fluency—not because activated L2 representations compete, but because L2 practice comes at the expense of more L1 use.

A problem with testing this prediction is how to obtain a pure measure of lexical knowledge in both English and the Other language that is not influenced by additional factors. To advance the argument, assume, for now, that self-ratings by bilinguals for general proficiency in English and their Other language provided reasonably valid measures of lexical knowledge (A study by Marian, Blumenfeld, and Kaushanskaya (2007) [4] correlated self-report measures of language proficiency with eight different measures of language skill covering both comprehension and production. For L2 (the proficiency of greatest concern in classifying an individual as bilingual), all 24 correlations between the three subjective measures and the eight objective measures were significant with r values ranging from 0.29 to 0.74 with a mean of 0.59. Taking all of their results into account, Marian et al. concluded that self-ratings are “an effective, efficient, valid, and reliable tool for assessing bilingual language status” (p. 960)). One could then ask if verbal fluency was determined solely by lexical knowledge in English or if it was also affected by lexical knowledge in the Other language. This can be examined by conducting a multiple regression with self-ratings for both English and the Other language as predictors and verbal fluency as the outcome variable. The amount of lexical knowledge in English should clearly be a potent predictor when the test language is English. The interesting question is whether lexical knowledge in the Other language predicts additional and previously unaccounted for variance in English verbal fluency. This is exactly what one would expect if the non-target language routinely interferes with the use of the target language. To foreshadow the results, this is what happens with the caveats described below.

Although bilingualism researchers routinely use self-ratings of proficiency in understanding, speaking, and reading [4] these self-ratings do not typically focus narrowly on vocabulary knowledge. To the degree that other aspects of “proficiency” (e.g., grammar, pragmatics, text/story comprehension, accent) may be taken into account by the rater and may differ in alignment (e.g., a bilingual might rate her L2 vocabulary knowledge as excellent, but her grammar as only adequate), a general proficiency scale may provide only an approximation to her true vocabulary knowledge. Thus, a clear limitation of past studies and the current study (As described below, we are using a pre-existing dataset and the primary purpose of the original study was to solicit ratings of general fluency in each language, not just vocabulary knowledge.) is that self-ratings of proficiency in each language are general ratings of speaking and understanding, not ratings narrowly focused on word knowledge. Thus, the specific research question tested in the current study is: Does self-rated proficiency in a non-target language account for unique variance when both English proficiency and Other Proficiency are used as predictors of English verbal fluency? If the answer is yes, this would be consistent with the interpretation that the Other language has an adverse effect during English verbal fluency. If the answer is no, then it would appear that once a single-language context has led to the selection of a target lexicon, then the selection of specific entries and their production can be carried out without interference from the non-target lexicon, as suggested by Costa and Caramazza [5].

## 2. Methods

The current study reports new analyses of an existing dataset and, consequently, many additional details are presented in the original study [1].

### 2.1. Participants

Participants in [1] were 236 San Francisco State University undergraduate students who earned extra credit in a psychology course for their participation. On the basis of self-ratings of proficiency in a language other than English, they were partitioned into 122 bilinguals and 108 monolinguals. In making their ratings, they were instructed to consider both comprehension and production of spoken language. In the original study, participants were classified as bilingual if their self-rated proficiency in a non-English language was at least 4.0 (can converse with little difficulty with a native speaker on most everyday topics, but with less fluency than a native speaker) on a 7-point scale. However, in order to examine a greater range of L2 fluency, the new analyses are based on the 144 students who rated their other language proficiency as at least 3.0 (can converse with a native speaker only on some topics and with some difficulty). The distributions of proficiency scores for English and the other language for the 144 bilinguals are shown in Table 1. As students were attending a university where the language of instruction is English, it is not surprising that most students rate their English as a 6 (as fluent as a native speaker) or a 7 (better than a typical native speaker). In contrast, as shown in the bottom row of Table 1, there is a more even distribution of Other Proficiencies across the scale values of 3 to 7. Operationally defined in terms of the ratio of English Proficiency to Other Proficiency, 61.8% are English dominant, 20.8% are balanced, and 17.4% are dominant in their non-English language. The majority of the bilinguals were sequential bilinguals, but 26.4% reported learning both languages simultaneously. For the subset of sequential bilinguals, the mean age-of-acquisition (defined as age when first exposed to a language) for English was 5.0 years and for their other language it was 2.3 years. Collectively, this group of bilinguals spoke 22 different languages, but 58% spoke Spanish, 20% Chinese, and 7% Tagalog. For our regression analyses, the languages must be partitioned into English and Other-than English because the verbal fluency task was conducted only in English. Across bilinguals, the Other language is sometimes a native language and sometimes a second language.

### 2.2. Verbal Fluency Tasks

In the verbal fluency tasks, participants were asked to produce as many words as they could in one minute that followed the prescribed rule. There were two trials of single-letter fluency (words beginning with F, words beginning with A) and two trials of single-category fluency (musical instruments, vegetables). The target categories were selected on the basis of previous work with the intent to have targets of comparable and intermediate difficulty. They were not explicitly told to respond in English, but all participants did so. Between trials, bilinguals would occasionally mention that they thought of the name of a vegetable or instrument in their other language, but did not know it or could not think of the word for it in English.

## 3. Results

### 3.1. Descriptive Statistics for Various Types of Verbal Fluency

For completeness, we show the mean number of correct responses, the number of intrusions from an incorrect category, and the mean number of repeated responses for each of four verbal fluency tasks given to each participant. Our analyses focus on the single-category and single-letter trials. Inspection of Table 2 confirms that the four targets (instruments, vegetables, “F”, and “A”) were of similar difficulty, each yielding an average of about 11 correct responses with standard deviations of about 4.

The distinction between category- and letter-fluency has been of theoretical interest because of the twin hypotheses that there is a bilingual advantage in executive functioning (EF) and that letter-fluency may require more EF. If both hypotheses were correct, then any bilingual disadvantage in category fluency might be reduced in the letter-fluency task because bilinguals could recruit their superior EF to close or eliminate the performance gap. Although some earlier studies reported a Group (bilingual versus monolingual) by Task (category fluency versus letter fluency) interaction with just such a pattern, Paap et al.’s review [1] showed that most studies reported additivity. Likewise, their large sample study yielded neither a main effect of Task nor a Group × Task interaction. This suggests that for the present purposes of testing for cross-language effects in bilinguals, one could average across the four single-target tasks and use that mean as a more stable composite measure. In an intermediate step to forming that composite, we compared the category mean (M=11.2, SD = 3.61) to the letter mean (M = 11.5, SD = 3.4) and those means did not differ, *t*(127) = 1.02, *p* = 0.311. Thus, in all subsequent analyses any reference to verbal fluency refers to the composite mean based on the two categories (instruments and vegetables) and the two letters (“F” and “A”).

### 3.2. Regression Analyses with Verbal Fluency as the Outcome Variable

The primary hypothesis of this study is that Other language proficiency can account for unique variance in an English-language test of verbal fluency when English language proficiency is included in the regression model. Table 3 shows the results of several regression models that include two or more predictors and with verbal fluency (tested in English) as the outcome variable.

*Model 1*. English Proficiency and Other Proficiency are the two predictors. Tested as a single bivariate correlation, it is not surprising that self-rated proficiency in English predicts the number of correct responses in a test of English verbal fluency, *r*(128) = +0.27, *p* = 0.001. In regression Model 1, the two predictors combined to account for about 16.4% of variance in the composite verbal fluency scores, *R* = 0.406, *p* < 0.001). The standardized coefficients showed that both were significant predictors, β = +0.18, *p* = 0.033 for English fluency and β = −0.32, *p* <0.001 for Other Proficiency. As predicted, the correlation for the test language (English) was positive, but negative for the competing language (Other). Observing a significant negative effect of Other Proficiency is consistent with the hypothesis that there is cross-language interference. Impressively, it is numerically more potent than proficiency in the test language. Given that English Proficiency and Other Proficiency are correlated with one another, *r*(128) = −0.27, *p* = 0.001, it is prudent to assess collinearity. Following the guidelines provided by Field [6], the VIF statistics are well below 10 and the tolerance statistics are well above 0.2.

*Model 2.* Many of our bilinguals (and especially those who speak Cantonese or other languages that do not use an alphabetic script) do not write or read their non-English language. In our past publications, we have considered proficiency in spoken language and have ignored reading. In this past work, if we had included reading and writing in our measure of other-language proficiency, then some native and highly fluent speakers would have mediocre proficiency scores even though they were native speakers and listeners. Thus, the analyses in the original paper [1] and in Model 1 measured both English and Other proficiency based on self-ratings of speaking and aural comprehension. However, our participants are college students and spend a lot of time reading English. In order to make sure that our Model 1 results were not biased by excluding reading, Model 2 included self-rated proficiency in reading English as an additional predictor. Reading proficiency in English does, unsurprisingly, correlate quite well with spoken language proficiency in English, *r*(125) = +0.57, *p* < 0.001. However, does Other Proficiency still account for unique variance in verbal fluency when both written and spoken proficiency are included in the model? Yes, self-related proficiency in the Other language accounts for unique variance in English verbal fluency even when both spoken English proficiency and reading English proficiency are included in the model, β= −0.31, *p* < 0.001. Thus, Model 2 (like Model 1) is consistent with the hypothesis that cross-language effects play a significant role in verbal fluency.

*Model 3*. Researchers in bilingual language control have emphasized the importance of several facets of current use and these include the relative use of each language and the frequency of language switching [1,7]. To the extent that these current experiences modulate the accessibility of each lexicon, they may play a role in verbal fluency. In Model 3, Percentage Use of English and Frequency-of-Language Switching are added to the original two predictors (English Proficiency and Other Proficiency) of Model 1. Neither of the current use predictors accounts for unique variability in verbal fluency: for Percent of English, β= +0.21, *p* = 0.117 and for Frequency-of-Language Switching, β= +0.22, *p* = 0.09. Again, the most important result is that Other Proficiency accounts for unique variance even when English Proficiency, percentage of current English use, and frequency-of-switching are taken into account, β= −0.35, *p* = 0.004.

*Model 4.* The original dataset includes another measure that is arguably more directly related to English vocabulary knowledge and that is the number of correct responses on the Multilingual Naming Task (MINT) [8]. Model 4 replaces self-ratings of English proficiency with MINT scores. Note first that Model 4 accounts for 38.2% of the variance in verbal fluency compared to only 16.4% in Model 1. The MINT predictor is the most potent of those examined so far, β= +0.56, *p* < 0.001. Furthermore, Other Proficiency does not account for any unique variance not accounted for by MINT, β= −0.10, *p* = 0.206.

*Model 5.* Model 4 potentially upsets the apple cart. It clearly calls into question the conclusion that the earlier models implicate cross-language interference as Other Proficiency does not account for unique variance when MINT scores are the other predictor. This might be expected if MINT scores were a more valid measure of vocabulary knowledge than self-ratings of general English proficiency. On the surface, this seems reasonable as self-ratings are often viewed as subjective, fuzzy, and open to bias. Furthermore, picture naming (MINT) is an objective measure that enjoys face validity as a measure of productive vocabulary. Perhaps the independent contribution of Other Proficiency observed with Models 1 to 3 was enabled by a mediocre measure of English vocabulary, namely, self-rated English Proficiency.

Although the preceding argument seems quite plausible, it fails to consider a potentially critical aspect of picture naming (MINT is, of course, a picture naming task), namely that picture naming may also be influenced by cross-language interference. If MINT performance already reflects cross-language interference, then there would be little or no unique variance for Other proficiency to account for when predicting verbal fluency. If this is the case, then Other Proficiency should strongly predict MINT scores when they trade places compared to Model 4. Thus, Model 5 used English Proficiency and Other Proficiency as predictors, but uses MINT scores as the outcome variable. The key result is that Other Proficiency accounts for unique variance in MINT scores, β= −0.33, *p* < 0.001.

## 4. Discussion

The first three regression models established that Other proficiency accounts for unique variance in English verbal fluency beyond that accounted for by English proficiency. This implicates a cross-language effect in verbal fluency in a single-language context. However, Model 4 replaced self-rated English proficiency with MINT English scores and the inclusion of Other proficiency no longer improved the model. If the outcome of Model 4 was simply due to MINT being a more reliable measure of English proficiency compared to self-ratings, then this would undermine the evidence for cross-language effects obtained in Models 1 to 3. Alternatively, we suggested that MINT may not be a more reliable measure, but rather one that is, itself, influenced by cross language interference. If MINT scores are a composite of benefits from greater English proficiency and cost from greater Other proficiency, then self-rated Other proficiency is less likely to account for additional unique variance in English verbal fluency. The coherence of this account rests on the assumption that self-rated English proficiency is less effected by Other proficiency (and hence Other proficiency can scoop up variance unaccounted for by English proficiency) but that MINT scores are influenced by Other proficiency. We consider next the plausibility of this account by speculating on how bilinguals self-rate their proficiencies and linking these speculations to work on self-ratings recently published by Tomoschuk and colleagues [9].

### 4.1. What Determines Self-Ratings of Language Proficiency?

The subjective experience of proficiency or fluency for a bilingual could rely on three different frames of reference. One framework for a bilingual is to compare fluency in one language to that of the other. Tomoschuk et al. tap into this type of comparison when they explicitly ask bilinguals to rate the degree to which one of their languages is stronger and conclude that these dominance ratings are more closely aligned with their MINT scores than numerically rating individual languages on a seven-point scale. Thus, in order to explain the difference between absolute ratings of proficiency and dominance ratings, one must conclude that self-ratings do not heavily rely on the relative comparison of fluency in one language compared to the other.

A second source of information about proficiency would be a bilingual’s perception of their proficiency relative to others. If the others are also bilinguals, the effects of cross-language interference may be neutralized. For example, the feeling that *“My dominant language is English and I am very fluent AND just as good as other bilinguals who have English as a native language”*. However, we know that, on average, there are bilingual disadvantages compared to monolinguals in verbal fluency and picture naming. Thus, self-reflecting on comparisons to other bilinguals will lead to an overestimation of proficiency compared to MINT. We suggest that the overestimate is caused by MINT’s sensitivity to cross-language effects and the absence of such effects when other bilinguals serve as the frame of reference for self-ratings. This agrees with Tomoschuk et al.‘s assertion that “*One possible explanation for this pattern of results is that different participants have different frames of reference that they use to evaluate their language proficiency*” p. 532.

Only if bilinguals have rich opportunities to compare their performance in their dominant language to the performance of monolingual speakers of that language would they have evidence that their proficiency was, on average, less than monolinguals. For example, from the perspective of a bilingual: *“My dominant language is Chinese and I am very fluent, but not quite as good as my peers who speak only Chinese.”* Considering this evidence should reduce the tendency to overestimate Chinese proficiency. In support of this argument, Tomoschuk et al. show striking differences between Chinese–English bilinguals raised in the USA compared to a group who immigrated to the USA relatively recently. The recently immigrated group (who had minimal exposure to English growing up) self-rated their Chinese fluency in closer alignment to their MINT scores, in other words, they had lower self-ratings. Tomoschuk et al. suggest that recently immigrated speakers are comparing themselves to family and friends in China, many of whom are native eiChinese speaking monolinguals who, by definition, have not been adversely influenced by acquiring or using an L2.

A conjecture offered here is that self-rated English proficiency in a bilingual is a weighted average of: (1) comparing my languages to each other, (2) comparing my proficiency to that of other bilinguals, and (3) comparing my proficiency to that of monolinguals. The first two will lead to an overestimation of MINT scores that are caused by failing to take adverse cross-language effects into account. This provides a valid interpretation for Model 1: Other proficiency predicts unique variance in English verbal fluency beyond that directly accounted for by English proficiency. This would not be the case if self-ratings of English proficiency fully reflected the adverse effects accruing from an L2. If so, there would be no unique variance in verbal fluency for L2 proficiency to account for. Because the evidence drawn from Tomoschuk et al. is somewhat circumstantial, we emphasize that we are speculating, not claiming, that the self-ratings of our bilinguals are dominated by adopting a frame-of-reference of other bilinguals. It may be worthwhile to explore the possibility that bilinguals could be instructed to use the three different frames of reference and generate reliability different self-ratings.

### 4.2. Interference

One account of the bilingual disadvantage in verbal fluency is cross-language interference. This is intuitive if one thinks of Model 1 as reflecting online causal relationships during a verbal fluency task. That is, what happens when a bilingual is asked to generate names of vegetables? At some point, attention may gravitate toward the concept for CUCUMBER, but the word form “pepino” may compete with the word form “cucumber” and render “cucumber” inaccessible. Thinking of Model 1 as an online model of verbal fluency, English Proficiency reflects the strength of English word forms and Other Proficiency the strength of competing words from the non-target lexicon. Thus, cross-language interference will increase as Other Proficiency increases.

### 4.3. Weaker Links

In contrast, Model 1 can be thought of in terms of the lexical structures that have been acquired over the bilingual’s lifetime and not-so-much as a description of online processing. This opens the door to explanations of the bilingual disadvantage in verbal fluency that appeal to weaker links rather than interference. Model 1 showed that Other Proficiency degrades verbal fluency. Although this could be due to interference, the “damage” could have been caused by past language experience. That is, if a bilingual is a native speaker of Spanish, acquires English when she starts school, and uses mostly Spanish at home and with friends, then she accrues far less experience in traversing links like the one from CUCUMBER to “cucumber” compared to a monolingual English-speaking peer. The weaker link alone may render “cucumber” inaccessible without suffering direct inhibition or suppression at the hands of “pepino”. The relationship between English Proficiency and Other Proficiency may reveal the importance of the history between the two measures rather than their interaction at the time of testing. That is, Other Proficiency is associated with lower verbal fluency not because it interferes more with finding correct English words, but because those individuals with higher proficiency, in say Spanish, have been practicing Spanish at the expense of English and, consequently, have forged weaker English links. Thus, although Model 1 shows that Other Proficiency accounts for unique variance in English verbal fluency apart from that directly accounted for by English Proficiency, there is nothing in the data considered to this point that rules out an explanation based on weaker links.

The most direct evidence supporting the weaker links interpretation is that the bilingual disadvantage in picture naming is greater when producing low-frequency than high-frequency picture names [10]. As nicely demonstrated in [10] (Table 1), it follows from prevailing assumptions about how experience influences connection strengths that frequency effects will be greater at lower levels of experience and diminish with increasing experience. Thus, the fact that frequency effects in picture naming are larger in bilinguals compared to monolinguals supports the weaker links hypothesis. Gollan et al. (2008) and Sandoval et al. (2018) view this as a diacritical test between weaker links and interference because they assume that cross-language interference makes the opposite prediction. That is, they assume that interference is restricted to the high-frequency range of words because only in this range will English words surely have an active competitor in the non-dominant language (e.g., Spanish). For example, “alchachofa” cannot interfere with “artichoke” if an English dominant bilingual does not know the Spanish word. This point about interference is true when the word has been acquired in only one lexicon. Furthermore, many low-frequency words are cognates because they are simply borrowed (e.g., many English monolinguals have acquired the words jicama and tomatillas) and cognates do not usually compete (but see [11] for some exceptions).

However, in contrast to the above, when translation equivalents (that are not cognates) exist in both lexicons, low-frequency words may be more vulnerable to interference than high-frequency words. For example, an English dominant bilingual growing up in the self-proclaimed artichoke capital of the world (Castroville, California) may know “alchachofa” but suffer intense interference because “artichoke” has very high frequency. All things considered, it is not clear (at least to us) that cross-language interference in picture naming (or verbal fluency) should be more prevalent for high-frequency picture names compared to low-frequency, especially when a non-target competitor enjoys a frequency advantage over the target. It appears that both weaker links and cross-language interference continue to compete as explanations for the bilingual disadvantage in verbal fluency. Of course, they could both contribute to the overall effect.

### 4.4. Interference in Green’s Inhibitory Control Model

Given its extraordinary influence in bilingualism research, it is worthwhile to consider our results in the context of Green’s Inhibitory Control Model (ICM) [12]. The ICM emphasizes that task schemas compete with each other for controlling action. Thus, depending on changes in topic or conversational partners, bilinguals may decide to switch from the “speak-English” schema to the “speak-Spanish” schema that reactively inhibits the English lexical representations via their language tags. Thus, switching languages triggers top-down inhibition of the previously-used language via the “language” nodes.

However, the ICM makes an interesting and more specific assumption, namely, that the actual inhibition is not applied until competition for production occurs. Consider a picture-naming task where a pre-cue indicates whether the upcoming picture should be named in English or Spanish. When the pre-cue requires a switch, say from Spanish to English, the new schema (“name-in-English”) must be selected and this sets the stage for the global inhibition of all the lemmas with Spanish tags. However, the actual inhibition is not applied until: (1) the picture appears, (2) the participant initiates plans to name the picture, and (3) response competition is detected. For example, if the stimulus is a picture of a CAT, then the competition between the correct response “cat” and the competitor “gato” will (4) trigger inhibition of all Spanish lemmas in proportion to the degree that they were activated by the picture. Thus, “cat” would be inhibited more than unrelated English words like “table”. In summary, Green’s ICM assumes that competition between the lexicons is controlled via inhibition and that inhibition is reactive, that is, in proportion to the amount of competing activation that needs to be neutralized.

What does the ICM predict for verbal fluency? If the task language is English and the memory search starts to hone in on a potential exemplar (e.g., CAT) then “cat” and “gato” will compete. If the task schema is “speak-English”, then all the Spanish lemmas will receive inhibition, but “gato” will receive the most because CAT has led to the coactivation of both “cat” and “gato”. However, the cross-language interference account of the bilingual disadvantage in verbal fluency relies on the inhibition flowing in the opposite direction, namely, from “gato” to “cat”. That is, might there be occasions when “gato” gets the jump on “cat” (despite the fact that the “speak-English” schema is selected) and suppresses “cat” to the point where it is inaccessible? Although it is easy to imagine that the initial competition delays the production of “cat”, it seems far less likely that “gato” can completely block the access of “cat”. Thus, we argue that the interference account of the bilingual disadvantage in verbal fluency is not very compelling within the framework of Green’s ICM.

### 4.5. Reflections on Inhibition as a General Construct

Inhibition is usually viewed as a domain-general ability to suppress competing or irrelevant information during goal-directed behavior. Recently, Rey-Mermet, Gade, and Oberauer [13] have urged us to *“stop thinking about inhibition”* in that way. They required each participant to complete six tasks assumed to reflect *Inhibition of Prepotent Responses* and five assumed to reflect *Resistance to Distraction* (hypothetically a different type of inhibition). Rather than examining only the fit of various models, Bayesian hypothesis testing was also conducted. These additional tests showed that the data are ambiguous as to: (a) whether there is one inhibition factor or two and (b) if there are two factors, whether they are correlated or not. Another problem pointed out by Rey-Mermet et al., both in their data and in other latent-variable analyses of the inhibition construct, is that for each latent variable, the loading for one task tends to dominate the others. Consequently, each latent variable represents mainly the variance of one task with the remaining tasks burdened with high error variances. These problems led Rey-Mermet et al. to suggest that all latent-variable models of the inhibition construct have poor explanatory power. From their view of this evidence, Rey-Mermet et al. suggested that nonverbal interference tasks do not measure a common underlying latent variable associated with general inhibitory-control, but rather a highly task-specific ability to resolve the type of conflict instantiated in each task.

If inhibition is task specific, as argued by Rey-Mermet et al. (2018), then there would be no basis for a bilingual advantage in domain-general inhibitory control (see [14] for a detailed examination of this evidence). Taking the critique a step further, MacLeod and colleagues [15], in an influential essay titled *In Opposition to Inhibition,* provided alternatives to inhibition for several classic demonstrations of “inhibition” including inhibition of return (IOR), Stroop interference, and negative priming. This begs the question: Is inhibition needed to account for bilingual language control?

In contrast to Green’s ICM Dijkstra and van Heuven’s [16], bilingual interactive activation plus model (BIA+) assumes no top-down inhibitory control. Recently, Dijkstra and colleagues [17] extended the computational version of BIA+ to production. The new model, called Multilink, successfully simulated many classic results obtained with picture naming and translation tasks. Critical to the present discussion, bilingual language control remains encapsulated within the word identification system and there are no inhibitory connections either within or between representational levels.

If inhibitory control, either specific to the word identification system or as a general construct, is placed on the chopping block, then weaker links emerge as the best explanation of the bilingual disadvantage in verbal fluency. Furthermore, the effectiveness of Other Proficiency as a predictor of verbal fluency should be considered as the historical effect of time spent speaking another language and, consequently, poaching practice time from the dominant language.

### 4.6. But Is There Not Other Evidence for Inhibition in Bilingual Language Control?

Neumann and colleagues have conducted and reported a series of elegant experiments [18,19,20] that yield a pattern of results that can be neatly explained by an account that embraces two types of inhibitory control. Their studies should stave off any march to judgment that inhibition is not involved in bilingual language control. However, after describing the most critical of Neumann’s experiments we will challenge the assertion that a non-inhibitory explanation, based on episodic retrieval, is undermined by these results. To presage the conclusion, it is too soon to either reject or accept the proposition that a domain-general inhibitory process plays an important role in cognitive control.

Many of Neumann’s experiments test for both positive and negative priming in couplets consisting of a priming event followed by a probe event. Each event consists of a target in lowercase and a distractor in uppercase. The distractor appears either above or below the target and should be ignored. Bilinguals make a naming response to the prime target and a lexical decision about the probe target. In the most relevant variant, naming (the prime event) is always in one language and the lexical decision (the probe event) in the other language.

The evidence for local inhibition accrues from the presence of negative priming (NP) and this reliably occurs when the prime event is in L1, the probe event in L2, and the prime distractor (e.g., CAT) becomes the probe target (viz., cat). Lexical decisions to cat are slower when CAT was the distractor compared to an unrelated control. To be more specific, in order for NP to occur (according to the interference account) CAT must be automatically activated despite the intention to ignore it. If CAT becomes sufficiently activated, then it will compete with the to-be-named target (e.g., box) and will trigger reactive inhibition of CAT to resolve the conflict. If the inhibition of CAT persists into the probe event and cat appears as the probe target, then the suppressed activation will result in a slower lexical decision compared to a control trial.

Although inhibition is an intuitively appealing explanation for NP, it is one of several phenomena (e.g., Stroop interference, inhibition-of-return) that MacLeod and colleagues [15] showed can be explained without having to evoke inhibitory control. In the case of NP, the most common alternative is Neill’s episodic retrieval account [21]. From this perspective, the appearance of a new stimulus elicits from memory the most recent event that includes a stimulus that is identical or similar. The retrieved event will result in positive priming when the previous stimulus was a target and negative priming if it was a distractor. Negative priming occurs because the probe stimulus leads to the retrieval of an event file consisting of the probe distractor paired with a “do not respond” tag. This follows from the assumption that it will take additional time to resolve the conflict between the correct response code and the retrieved code.

The critical contrast in Neumann’s experiments is that NP occurs when the consistent order of tasks is L1 (naming) then L2 (lexical decision), but not the reverse. This is what would be predicted from the interference perspective if inhibition is reactive and in proportion to need. If an L1 distractor is harder to ignore (more automatic) than an L2 distractor and is more highly activated, then it will receive more inhibition. If the inhibition actually drives the activation of the distractor below the normal baseline, then it will take longer to make the lexical decision when that distractor becomes the probe target. Indeed, NP occurs in the L1-then-L2 order. However, if the order is L2-then-L1 the amount of inhibition applied to the distractor in the initial naming task should be less because less is needed. If the amount is insufficient to carry over into the probe task, then significant NP may not occur.

This interference account of NP is usually presented as intuitive and fully following from Green’s ICM. As an aside, we suggest that in the absence of a computational model, there are implicit parameter decisions that are needed to predict the L1/L2 asymmetry and these may not enjoy high plausibility. To illustrate, consider the assumption that the amount of inhibition is in proportion to need. If this could be precisely titrated, then “need” might be defined as the amount of inhibition needed to return the distractor to the normal baseline. If L1 distractors are driven back down to baseline by substantial inhibition and L2 distractors driven to baseline by more moderate amount of inhibition, then it would not matter if the prime task used L1 or L2. In fact, there would be no negative priming because L1 distractors, L2 distractors, and control distractors would all be at baseline at the start of the probe trial. Thus, we suggest that an interference account of NP requires that inhibition is applied proportionately and beyond what is needed such that L1 distractors are driven below normal baseline and L2 distractors just to baseline. Thus, the interference account of NP may be more of a “just so” story than generally acknowledged.

Neumann and colleagues assert that their between-language findings “undermine” episodic retrieval models, in part, because they do not predict the L1/L2 asymmetry in NP effects. This is true if you restrict the episodic retrieval models to a single parameter specifying whether the response codes either match or mismatch. However, just like interference models, there are many implicit assumptions that could be instantiated in different versions of a computational model. Consider for example the formation of an event file consisting of a stimulus and the response to that specific stimulus. What governs the probability of encoding and storing an episode or the strength of that memory trace? The distractors are located just above or below the targets and are likely to be encoded even if the participant’s intention is to ignore them. However, it would seem reasonable to propose that the likelihood or strength of distractor activation and its episodic trace would vary as a function of its salience and its propensity for automatic coding. In this framework, L1 distractors should be advantaged compared to L2 distractors just as L1 words generate more Stroop interference compared to L2 words. We suggest that it is not easy to judge which theory has the most parsimonious explanation for the observation that NP occurs in the L1-then-L2 order, but not the reverse.

Another aspect of Neumann’s cross-language experiments is that the L2-then-L1 order produces positive priming (PP), but not NP. Based on translation priming in simpler tasks not involving distractors, PP would not have been predicted. As Grainger and colleagues [22] pointed out, translation priming in proficient bilinguals is *“a surprisingly elusive phenomenon”* (p. 274) when primes are in the L2 and targets are in the L1 (see also [23]).

Neumann and colleagues do not dwell much on why PP occurs in the L2-then-L1 order for their task, but rather seek to explain why PP disappears in the L1-then-L2 order. Their account invokes a second type of inhibitory control, namely, proactive suppression of the language used in the prime task. This follows from the assumption that the participant knows that the probe task will always be in the other language and that an excellent strategy would be, at the onset of the LD task, to globally inhibit the language used in the naming task. However, if complete and global suppression of the previously used language was possible, why would participants not engage in proactive inhibition in the L2-then-L1 order as well? If proactive inhibition was effortful, it would even require less effort to fully suppress L2.

These questions aside and switching to the other perspective, it is clear that episodic retrieval models would not predict the absence of PP in the L1-then-L2 order based on the simple notion of response codes that either match or mismatch because they do match. However, perhaps it is the presence of PP in the L2-then-L1 order that deserves additional scrutiny given its rarity in other translation priming tasks. The fact that the probe task is LD may impose limits on the generality of the findings because the LD task is rather infamous for its vulnerability to strategic influences on the criteria used to classify the target string as a word or nonword [24,25].

In summary, the series of experiments by Neumann and colleagues offers a tantalizing pattern of results that follow from one instantiation of an interference account of bilingual language control. However, caution is recommended before assuming that the “inhibitory control” reflected in NP is an instance of domain-general inhibitory control. An individual differences examination would be useful [26,27]. We know that interference scores in the flanker and Simon task do not correlate with one another. Indeed, even the two classic versions of the flanker task (arrows or letters) do not correlate with one another. A similar approach to NP would be to correlate the magnitude of NP in one task with that obtained in another. If NP reflects a domain-general inhibitory process, they should show significant correlations. To close the loop, would the magnitude of NP correlate with flanker, Simon, or Stroop effects?

### 4.7. Would Measures of Inhibitory Control Predict Verbal Fluency?

An individual differences approach could also be used in a replication of the current study by including a measure of inhibitory control and then correlating that measure with verbal fluency. If a domain-general measure of inhibitory control predicted verbal fluency, this would support the interpretation that part of the bilingual disadvantage in verbal fluency was caused by interference from the other language. This is a challenging recommendation given the controversy regarding the construct of inhibition and claims that there may not be a domain-general inhibitory ability. Be that as it may, our original study did include three different cued switching tasks and the switch costs from those tasks correlated with each other, thus demonstrating convergent validity. Furthermore, switch costs are generally assumed to be caused, at least in part, by having to inhibit the task schema active on the previous trial on “switch trials” in contrast to “repeat trials”. In order to explore possible associations between measures of inhibitory control and verbal fluency, we correlated the switch costs in the color-shape switching task with total verbal fluency, but the correlation was not significant, *r*(122) = −0.072, *p* = 0.43. Thus, there is no evidence from this test that bilinguals with better inhibitory control (and presumably better able to inhibit the interference from the Other language) have better verbal fluency as one would expect if cross-language effects in verbal fluency were, indeed, caused by interference. Furthermore, it is worth noting that switch costs did significantly correlate with MINT scores, r(128) = −0.223, *p* = 0.01, but not with self-rated English proficiency, *r*(138) = −0.043, *p* = 0.62. This pattern is consistent with the hypotheses that inhibitory control affects performance in MINT, but does not influence self-ratings of English proficiency. These types of relationships could be productively studied in future work.

## 5. Summary and Conclusions

There are small but fairly consistent bilingual disadvantages in picture naming and tests of verbal fluency. These disadvantages could be due to cross-language interference or weaker-links between semantic concepts and their phonological form. Both hypotheses lead to the prediction that the degree of proficiency in the nontarget language should be negatively correlated with and account for unique variance in verbal fluency performance apart from the direct effects of degree of proficiency in the target language. Although several regression models were presented that are consistent with this prediction, there are potential complications having to do with the measures of vocabulary knowledge and the degree to which different measures (e.g., self-ratings versus objective measures of receptive or productive vocabulary) are sensitive to adverse cross-language effects. In summary, the regression models showing that Other Proficiency is a potent predictor of verbal fluency cannot rule out the alternative explanation that the effect of Other Proficiency is simply due to past language usage that has led to weaker links in the test language.

Considering explanations of the bilingual disadvantage in picture naming and verbal fluency that do not involve inhibition led to a general discussion of recent claims that inhibitory control may not be a domain-general cognitive ability and, even more radically, that inhibition may play no role in cognitive control (see [20] for a more detailed discussion). If this is true, then during production (or comprehension) the non-target language may not influence the online processing in the target language. Rather, the influence of one language on the other may simply reflect the historical consequences of using one language at the “expense” of the other with degree of use determining the quality of the lexical representations of each language.

## Figures and Tables

**Table 1 brainsci-09-00175-t001:** Frequency distributions for English and “other” language proficiencies.

	Self-Rating of Proficiency in Speaking and Listening				
	3	4	5	6	7
English	0	8	14	46	76
Other Language	22	25	29	39	29

**Table 2 brainsci-09-00175-t002:** Mean number of correct responses, intrusions, and repetitions in each of the verbal fluency tasks (standard deviation in parentheses) for the bilinguals analyzed in the present study.

Task/Target	Correct	Intrusions	Repetitions
Single Category			
Instruments	11.6 (4.4)	0.20 (0.62)	0.16 (0.39)
Vegetables	10.9 (4.0)	0.23 (0.75)	0.10 (0.30)
Single Letter			
F	12.2 (3.7)	0.23 (0.58)	0.09 (0.32)
A	10.8 (3.9)	0.13 (0.35)	0.18 (0.58)

**Table 3 brainsci-09-00175-t003:** Regression models predicting verbal fluency or scores on the Multilanguage Naming Task (MINT) as indicated.

Regression Model Predictors	Outcome Variable	β	*t*	*p*	Zero−Order	Tolerance	VIF
Model 1 (*n* = 128)	Verbal Fluency						
English Proficiency		+0.18	+20.16	0.037	+0.27	0.93	10.08
Other Proficiency		−0.32	−30.72	<0.001	−0.36	0.93	10.08
Model 2 (*n* = 124)	Verbal Fluency						
English Proficiency		+0.10	+00.96	0.340	+0.29	0.62	10.60
Other Proficiency		−0.31	−30.65	<0.001	−0.36	0.92	10.08
English Reading		+0.18	+10.76	0.081	+0.28	0.66	10.57
Model 3 (*n* = 110)	Verbal Fluency						
English Proficiency		+0.17	+10.77	0.081	+0.30	0.84	10.20
Other Proficiency		−0.35	−20.94	0.004	−0.39	0.53	10.88
Percentage English Use		+0.21	+10.58	0.117	+0.34	0.44	20.29
Frequency of Switching		+0.22	+10.69	0.094	+0.19	0.46	20.19
Model 4 (*n* = 126)	Verbal Fluency						
English MINT		+0.56	+70.10	<0.001	+0.61	0.79	10.26
Other Proficiency		−0.10	−10.27	0.206	−0.36	0.79	10.26
Model 5 (*n* = 126)	MINT						
English Proficiency		+0.45	+60.23	<0.001	+0.55	0.92	10.08
Other Proficiency		−0.33	−40.51	<0.001	−0.46	0.92	10.08

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
