# Peer review of "Other Language Proficiency Predicts Unique Variance in Verbal Fluency Not Accounted for Directly by Target Language Proficiency: Cross-Language Interference?"

_brainsci, 2019, doi:10.3390/brainsci9080175_

Round 1

Reviewer 1 Report

I believe the authors have adequately addressed the issues I raised in my first review and the MS in my view is now much improved and ready for publication, after the new typos are fixed.  

I do have one quibble, however, in the red text section on line 529 there is a reference to the work of Schneider & Shiffrin (1977) with a comment about their "inconsistent mapping" condition that meant to apply to the studies under discussion in that section in some way.  This reference is inappropriate, if I remember their work correctly, because they used hundreds (or thousands?) of trials in which a couple of letters were always targets in the presence of a couple of distractor letters that were always the nontargets in the consistent mapping condition, and these same letters switched roles from trial to trial in the inconsistent mapping condition.  The main finding was that after hundreds and hundreds of trials of the consistent mapping condition, it was really hard for people to learn to look for and process those nontarget letters if the roles changed regarding what was target and what was nontarget information. There was no such problem finding as the currently designated  target letters after the inconsistent mapping condition.  I'm probably muddling this study up, because my memory of it is so vague, sorry about that. But I'd strongly recommend the authors go back and read that study again, because I'm pretty sure that there is not even the remotest connection between that study in the context of what is being discussed in that section. For one thing, the priming experiment authors presented words once and only once in the entire experiment, except in their attended repetition or ignored repetition conditions in which words would be presented a total of twice in the entire experiment. No consistent or inconsistent prior mapping/learning was involved in those experiments.  What, if anything, do they have to do with Shiffrin and Schneider's inconsistent mapping manipulation or the implications from those findings?

Author Response

We appreciate your acknowledgement that the manuscript is much improved and ready for publication.  Your "quibble" regarding inconsistent mapping and the Shiffrin & Schneider article was completely justified.  It is not relevant and we simply deleted this material.  

Reviewer 2 Report

I thank the authors for addressing some of my previous comments in the revision. However, most of the points I previously raised could not be addressed without designing and running another experiment. My following comments therefore still stand:

- This study does not do much to disentangle the two theories that have been proposed to explain lexical bilingual disadvantages: interference/inhibition vs weaker links due to less language use.

- The effects of 'other language proficiency' are still shaky. The authors now provide more of a discussion as to what self-rated proficiency can reflect as compared to objective proficiency scores such as the MINT. However, this revised discussion is very speculative (as the authors acknowledge, but it therefore also does not really help to interpret the data, simply because we do not know how the bilinguals in this study answered the self-rated proficiency questions and what reference framework they had in mind).

- A very large part of the paper is dedicated to questioning domain-general inhibition (now with the addition of commenting on work suggested by another reviewer). Given that the current data cannot say much about inhibition, I still feel like the manuscript consists of two separate parts that are not linked that closely.

I personally would advise the authors to either turn this manuscript into a review paper on domain-general inhibition (in line with their discussion on this topic) OR to present it as an experimental paper, in which case it would need additional experiments in my opinion to A) clarify why effects of other proficiency where significant when including self-rated English proficiency but not when including MINT scores, and/or B) to try and design a study that could say more about the underlying mechanisms (inhibition vs weaker links).

Author Response

In our introduction we state that: “…. the specific research question tested in the current study is: Does self-rated proficiency in a non-target language account for unique variance when both English proficiency and Other Proficiency are used as predictors of English verbal fluency?”  This is an empirical question and our data strongly support that the answer is yes.  The results clearly implicate an adverse cross-language interaction during the verbal fluency task.  The study was not designed to adjudicate the theoretical question of whether that negative effect of Other proficiency on English verbal fluency is caused by inhibition, weaker links, or some combination of the two.  Because the inhibition/interference explanation seems more intuitive I think one contribution of our discussion is to clearly develop an account of how weaker links could account for the outcome of our regression models.  Likewise we raise (I believe we are the first to do so) the question of whether self-rating of proficiency in one language are influenced by perceived proficiency in a bilingual’s other   and the parallel question of whether other proficiency influences objective performance in picture naming tasks such as the MINT.  Answering these questions would probably require a series of new experiments that extend far beyond our original research question as articulated above.  The debate between inhibition/interference and weaker links has been percolating for more than a decade and it is unlikely to be decided by any new single study. 

Reviewer 2 suggests that we might recast our submission into a review paper on the question of domain-general inhibitory control.  The inhibition/interference account of the bilingual disadvantage in verbal fluency critically assumes that there is such an ability.  However, the breadth of such a review paper would be great and the literature on the bilingual disadvantage in verbal fluency would constitute a very small proportion of the relevant literature.  Such a review paper would not provide a good fit to a special topics issue focusing on cross-language interaction in bilinguals.   In contrast, the current version of the manuscript provides an excellent fit.  Our revision did add an extensive section on Neumann’s negative priming studies, but given that Neumann’s paradigm actually involves language switching it seems more directly relevant to the inhibition versus weaker links debate and, in fact, we used it to frame our final summary and conclusions.  The alternative explanation we offer for Neumann’s results based on more than a single binary (match or no match of the response codes) parameter is novel and we think it will promote more careful analyses of whether the phenomenon of negative priming is really caused by inhibition of the distractor turned target.   Based on the extensive revisions we made to the original manuscript and Reviewer 1’s ringing endorsement of the revision, we think we have an interesting and informative contribution to make to this special issue and hope the editor and reviewers agree. 

This manuscript is a resubmission of an earlier submission. The following is a list of the peer review reports and author responses from that submission.

Round 1

Reviewer 1 Report

Review for: Brain Sciences

Manuscript ID: brainsci-518102

Title: Other Language Proficiency Predicts Unique Variance in Verbal Fluency Not Accounted for Directly by Target Language Proficiency: Cross-Language Interference?

Summary

The authors investigated cross-language effects using a combination of verbal fluency tasks whereby bilingual subjects name as many exemplars of a target category (e.g., vegetables) or words starting with a particular letter (e.g., F) within one minute. Multiple regression models were used employing predictors such as self-rated proficiency in English and the subjects’ other language in an effort to measure productive vocabulary. The findings were interpreted as showing that proficiency in the other language accounted for significant unique variance that was not directly accounted for by proficiency in English. A finding that was deemed consistent with two different accounts: 1) cross-language interference; and 2) bilingual disadvantages in verbal fluency are due to bilinguals having weaker links between semantic concepts and their phonological form. A broad framework is used in the discussion section to question whether domain-free inhibition has an important role in bilingual language control and whether such domain-general inhibition even exists.

Evaluation and Recommendation

On the positive side, the article is well written and addresses relevant issues in the bilingual literature. In particular, the present work may help to fill in some gaps in the current understanding of influences of a native language on the second language and vice versa regarding why bilinguals might show disadvantages in verbal fluency in each of their languages. On the negative side, however, in their discussion section the authors do not address recent research findings that challenge their rejection of domain-general inhibition having a role in bilingual language control. My concerns about this important point are outlined in the section below. If this major concern can be successfully addressed in a revision, the article should be considered for publication in Brain Sciences.

Major Concern

Most tasks used in bilingual research (including those of the present MS) are inadequate for tracking the role of domain-free inhibition in bilingual language modulation (see e.g., Li et al., 2017; Nkrumah & Neumann, 2018). It is therefore presumptuous to dismiss such a role on the basis of findings in the current MS given that such modulation occurs almost instantly, but then rapidly dissipates (see further key references below). In this slew of recent articles, within and cross-language target-to-target (attended repetition) and distractor-to-target (ignored repetition) priming effects were captured using  moment-to-moment processes under selective attention manipulations. They argue that in order to capture the role of inhibitory processing at either the level of a language, or individual words within a language, it is necessary to track such inhibitory effects online virtually from moment-to-moment. Taken together, those articles build the case (based on a consensus in the bilingual literature) that for proficient bilinguals there is parallel activation in both languages whenever a concept in one of their languages becomes activated. Hence, there is always a component of selective attention involved in accessing the word in the momentary target language, since only a subset of the information at hand is required to achieve that goal. Crucially, those authors make a compelling case that tracking both positive (facilitation) and negative (impairment) reaction time priming effects between languages with a selective attention component may be the only way to accurately gather evidence for the potential role of domain-free inhibitory processes in bilingual language modulation. A more nuanced view of bilingual inhibitory processing emerges from those articles which goes beyond the ICM and BIA+ perspectives covered in the present MS.

For the reasons outlined above, the experimental data used in the current MS are clearly inadequate to isolate domain-free inhibitory processing. It is therefore premature to dismiss inhibition as a domain-general ability to suppress competing or momentarily irrelevant information during goal-directed behaviour. Placing such a role in bilingual language modulation on the chopping block could be akin to throwing out the baby with the bathwater.

References:

Li, L., Neumann, E., & Chen, Z. (2017). Identity and semantic negative priming in rapid

       serial visual presentation streams. Attention, Perception, & Psychophysics, 79,

       1755-1776.

Neumann, E. & Nkrumah, I. K. (2019). Reversal of typical processing dynamics in

       positive and negative priming using a non-dominant to dominant cross-language

       lexical manipulation. Memory, 27(6), 829-840.  

Neumann, E., Nkrumah, I. K., & Chen, Z. (2018).  Excitatory and inhibitory priming by

       attended and ignored non-recycled words with monolinguals and bilinguals.

       Memory, 1-12.

Nkrumah, I. K. & Neumann, E. (2018). Cross-language negative priming remains

      intact, while positive priming disappears: Evidence for two sources of selective 

inhibition. Journal of Cognitive Psychology, 30, 361-384.

Minor Concerns

Line 119  the acronym EF is used but not defined until the next page (line 136).  Acronyms should be defined in the first instance in which they appear.

Line 154  similarly MINT is not defined until later (line 195), instead of when first appeared.

Lines 219-224  The first paragraph of the discussion section is hard to understand, and doesn’t the 3rd sentence in that paragraph contradict the first 2? Perhaps this paragraph can be clarified.

Lines 238-246   Maybe I’m missing something, but parts of this paragraph seem to contradict what is stated in paragraph 1 of the discussion (lines 2-19 224). Clarification seems warranted.

Line 335  shouldn’t the word “advantage’ be “disadvantage” instead?

Author Response

Response to Reviewer 1 Comments

Point 1: On the positive side, the article is well written and addresses relevant issues in the bilingual literature. In particular, the present work may help to fill in some gaps in the current understanding of influences of a native language on the second language and vice versa regarding why bilinguals might show disadvantages in verbal fluency in each of their languages. On the negative side, however, in their discussion section the authors do not address recent research findings that challenge their rejection of domain-general inhibition having a role in bilingual language control. My concerns about this important point are outlined in the section below. If this major concern can be successfully addressed in a revision, the article should be considered for publication in Brain Sciences.

Response 1: We appreciate the acknowledgement that our article is well written and fills some gaps in our understanding of cross-language effects.  We have added an extensive discussion of the recent research findings cited by Reviewer 1 that appear to implicate domain-general inhibition during bilingual language control. 

Point 2: Most tasks used in bilingual research (including those of the present MS) are inadequate for tracking the role of domain-free inhibition in bilingual language modulation (see e.g., Li et al., 2017; Nkrumah & Neumann, 2018). It is therefore presumptuous to dismiss such a role on the basis of findings in the current MS given that such modulation occurs almost instantly, but then rapidly dissipates (see further key references below).

Response 2: We did not intend to question the role of domain-free inhibition in bilingual language control on the basis of our regression analyses that include English and Other language proficiencies as predictors of verbal fluency.  Rather, we attempted to describe how a weaker-links account that does not require inhibition might provide an alternative explanation for our findings.  I think we are quite clear that there is nothing in our findings that leads to a rejection of an interference model.  However, in our broader discussion we appeal to a large array of recent articles that report serious problems in convergent validity and the inability to establish the concept of domain-general inhibitory control through latent variable analyses.  This is the basis for being skeptical of “inhibition”.  In our revision we argue for caution in appealing to domain-general inhibition, not for its immediate rejection. 

Point 3. In this slew of recent articles, within and cross-language target-to-target (attended repetition) and distractor-to-target (ignored repetition) priming effects were captured using  moment-to-moment processes under selective attention manipulations. They argue that in order to capture the role of inhibitory processing at either the level of a language, or individual words within a language, it is necessary to track such inhibitory effects online virtually from moment-to-moment. Taken together, those articles build the case (based on a consensus in the bilingual literature) that for proficient bilinguals there is parallel activation in both languages whenever a concept in one of their languages becomes activated. Hence, there is always a component of selective attention involved in accessing the word in the momentary target language, since only a subset of the information at hand is required to achieve that goal. Crucially, those authors make a compelling case that tracking both positive (facilitation) and negative (impairment) reaction time priming effects between languages with a selective attention component may be the only way to accurately gather evidence for the potential role of domain-free inhibitory processes in bilingual language modulation. A more nuanced view of bilingual inhibitory processing emerges from those articles which goes beyond the ICM and BIA+ perspectives covered in the present MS.  For the reasons outlined above, the experimental data used in the current MS are clearly inadequate to isolate domain-free inhibitory processing. It is therefore premature to dismiss inhibition as a domain-general ability to suppress competing or momentarily irrelevant information during goal-directed behavior. Placing such a role in bilingual language modulation on the chopping block could be akin to throwing out the baby with the bathwater.

Response 3: We are not convinced that the negative priming paradigm used by Neumann and colleagues enables one to track inhibitory effects online virtually from moment-to-moment.  Wouldn’t that require some continuous measure of the task relevant neural activity?  Be that what it may, these behavioral studies present a formidable package and we have added an extensive section to the discussion that describes them.  As impressed as we are by this work, the reviewer will discover that we have some questions about the implicit assumptions underlying the interference account and also suggest that the absence of NP in the L2-to-L1 order may not (as claimed) undermine the episodic retrieval account. 

Point 4.  Minor Concerns

 Line 119  the acronym EF is used but not defined until the next page (line 136).  Acronyms should be defined in the first instance in which they appear.

Response:  No longer a problem because the paragraph has been deleted. 

Line 154  similarly MINT is not defined until later (line 195), instead of when first appeared.

Response:  Fixed

Lines 219-224  The first paragraph of the discussion section is hard to understand, and doesn’t the 3rd sentence in that paragraph contradict the first 2? Perhaps this paragraph can be clarified.

Response:  This paragraph has been replaced and I hope the new version clarifies the argument. 

Lines 238-246   Maybe I’m missing something, but parts of this paragraph seem to contradict what is stated in paragraph 1 of the discussion (lines 219-224). Clarification seems warranted.

These paragraphs have been rewritten.  A key to our argument is the assumption that self-ratings are less influenced by cross-language interference, not that the influence is zero.  Perhaps this was, at least in part, the cause of the apparent contradiction.  

Line 335  shouldn’t the word “advantage’ be “disadvantage” instead?

Yes! Good catch. 

Reviewer 2 Report

This paper presents data from a published study to address a different question which wasn’t directly investigated in the original (published) article. The manuscript is concise and to the point. However, I do have a few comments and concerns, which I detail below. 

Generally, the language profiles of the participants are not really clear. Where they all native English speakers? Did they all have English as their L2? I think the terminology used in the manuscript is contributing to the confusion, rather than clarifying. Why call it Other language? Why not simply use “L1” and “L2”? Or is it because the authors have tested a large and heterogeneous group of bilinguals (where English is the L1 for some but the L2 for others), some of which have fairly low proficiency in one of their languages (as reported in Table 1). I’m sceptical about combining so many subgroups of participants into one group and drawing general conclusions about all bilinguals, especially in light of studies that have found differences between different “types” of bilinguals, divided by proficiency, age of acquisition etc. 

A big chunk of the introduction and rationale is based on the assumption that bilinguals’ “non-target lexicon interferes with their retrieval” of their L1. However, there are several studies looking at lexical selection in bilinguals that have in fact found no interference but instead facilitation from bilinguals’ L1 on their L2 retrieval (see Costa & Caramazza, 1999; Costa, Miozzo & Caramazza, 1999; Dylman & Barry, 2018; Higby, Donnelly, Yoon & Obler, in press). This significant and increasing finding should perhaps be considered in the current paper. 

Mainly using self-rated language proficiency as a predictor of verbal fluency seems a bit crude of a measurement. It would be more compelling if objective proficiency measures, perhaps vocabulary size measures, were included. To their credit, the authors do include MINT, but the statistical journey that follows (in Model 4, when self-reported English proficiency was substituted for MINT, other language proficiency no longer significantly predicted verbal fluency in English. This seems to be at odds with the authors’ general line of argument. The authors argue that MINT may “already reflect cross-language interference”, which they back up with Model 5, where other proficiency significantly predicts MINT scores) makes it seem just a bit shaky. Arguing that MINT reflects cross-language interference because MINT scores can be predicted by other proficiency seems a bit speculative. There are a number of reasons why other proficiency should predict MINT scores. Also, MINT is only a picture-naming task and neither a pure vocabulary size test nor a language proficiency test as such. Furthermore, the MINT scores are for naming in English only, is that correct? It’s therefore not an objective test of other proficiency. 

A few minor points

Lines 49-57. This explanation of the rationale is a bit confusing. Can the authors clarify a bit? 

Line 107. “...and for their other language [age of acquisition] was 2.3 years”. I’m confused. Wasn’t this other language this subgroups’ native language? Why didn’t they start acquiring it until the age of 2.3 years? What did they do before that?

Author Response

Response to Reviewer 2 Comments

Point 1:  Generally, the language profiles of the participants are not really clear. Where they all native English speakers? 

Response 1:  No.  Of the 122 bilinguals in Paap et al.(2017) 37 were native speakers of two languages.  For the sequential bilinguals 13 acquired English as their native language while 72 were native speakers of a language other than English.

Point 2:  Did they all have English as their L2?

Response 2:  No. 

Point 3:  I think the terminology used in the manuscript is contributing to the confusion, rather than clarifying. Why call it Other language? Why not simply use “L1” and “L2”? 

Response 3.  The terminology is completely consistent with the design of the study in which all participants completed both (letter and category) verbal fluency tasks in English.  The purpose was to test for cross-language effects on verbal fluency by using the self-rated proficiency in each of the two languages as predictors in a regression model.  Collectively the bilinguals spoke 21 different “other” languages.  Spanish (52%), Chinese (16%), and Tagalog (8%) were the most highly represented.  SFSU is primarily a commuter school and most of the bilinguals come from families living in bilingual communities with many local speakers of the same two languages.  We could not do similar analyses of L1 and L2 because we only have the outcome variable (total correct responses on the verbal fluency tasks) in English. 

Point 4:  Or is it because the authors have tested a large and heterogeneous group of bilinguals (where English is the L1 for some but the L2 for others), some of which have fairly low proficiency in one of their languages (as reported in Table 1). I’m skeptical about combining so many subgroups of participants into one group and drawing general conclusions about all bilinguals, especially in light of studies that have found differences between different “types” of bilinguals, divided by proficiency, age of acquisition etc. 

Response 4.  We think a more nuanced understanding of how the participants serve the research question and design would be helpful.   First, it is desirable for the bilingual participants to have fairly high levels of English proficiency so that there is plenty of room to see adverse cross-language effects.  The vast majority rate their English proficiency “as good as a native speaker” (6) or “better than a typical native speaker (7).   Most crucial, we are interested in the effects of the Other language on English verbal fluency and a good range of Other proficiencies is desirable (not undesirable) so that one can see increasing negative effects on English verbal fluency as Other proficiency increases.  One would have little or no chance of observing such a relationship if all the bilinguals were also highly fluent in their Other language.  This is not a between group design where homogeneity within each cell is desired.  Reviewer 2 makes a perfectly good point that other aspects of bilingualism may also be affecting the verbal fluency scores.  That is why we included % of English Use and Frequency-of-Switching as additional predictors in Model 3.  Given that Reviewer 2 explicitly mentioned age-of-acquisition I just added A-o-A of the Other language into a new version of Model 1 and A-o-A does not account for any of the unique variance in English verbal fluency. 

Point 5:  A big chunk of the introduction and rationale is based on the assumption that bilinguals’ “non-target lexicon interferes with their retrieval” of their L1. However, there are several studies looking at lexical selection in bilinguals that have in fact found no interference but instead facilitation from bilinguals’ L1 on their L2 retrieval (see Costa & Caramazza, 1999; Costa, Miozzo & Caramazza, 1999; Dylman & Barry, 2018; Higby, Donnelly, Yoon & Obler, in press). This significant and increasing finding should perhaps be considered in the current paper. 

Response 5.  What our regression models show is that proficiency in the Other language has an adverse effect on English verbal fluency.  It was not our intent to introduce our study with the presumption that the mechanism is inhibition rather than weaker links and have made some edits that emphasize that the adverse effects in our data could be due to interference, weaker links, or a combination of the two.  The important early work of Costa and Caramazza that proposes language selection without inhibition is now cited.  Dylman & Barry use the picture-word interference task and show that “distractors” that are translation equivalents, unlike semantically-related “distractors”, can produce facilitation.  Dylman & Barry’s preferred explanation is that there are excitatory links between translation equivalents that are stronger in the L1 to L2 direction..  This is interesting data and an interesting proposed explanation, but it doesn’t seem to constrain answers to the question of how does non-target proficiency adversely affect a test of English verbal fluency. Rather Dylman & Barry simply make the point that the association between Other proficiency and English proficiency is not always negative.  One might turn the question around and ask how Dylman & Barry’s conclusion that there are excitatory connections between translation equivalents account for negative priming in Neumann’s paradigm and negative regression correlations in our data. 

Point 6:  Mainly using self-rated language proficiency as a predictor of verbal fluency seems a bit crude of a measurement. It would be more compelling if objective proficiency measures, perhaps vocabulary size measures, were included. To their credit, the authors do include MINT, but the statistical journey that follows (in Model 4, when self-reported English proficiency was substituted for MINT, other language proficiency no longer significantly predicted verbal fluency in English. This seems to be at odds with the authors’ general line of argument. The authors argue that MINT may “already reflect cross-language interference”, which they back up with Model 5, where other proficiency significantly predicts MINT scores) makes it seem just a bit shaky. Arguing that MINT reflects cross-language interference because MINT scores can be predicted by other proficiency seems a bit speculative. There are a number of reasons why other proficiency should predict MINT scores. Also, MINT is only a picture-naming task and neither a pure vocabulary size test nor a language proficiency test as such. Furthermore, the MINT scores are for naming in English only, is that correct? It’s therefore not an objective test of other proficiency. 

Response 6.  Yes, the data we are analyzing from Paap et al. (2017) only have MINT scores for English because many of the bilinguals spoke a language other-than-English that has not been validated for MINT.  We wish we had an objective measure of Other proficiency, but we don’t.   We can only reiterate that Other proficiency did predict English MINT scores and that this is exactly what one would expect IF MINT scores were influenced by Other proficiency.  We hope that the material we have added from the Tomoschuk et al. (2019) investigation of self-ratings across different language groups may make Reviewer 2 feel somewhat less shaken.  On our part, we can only make certain that our strong assertions are in valid places.  We can say with confidence that self-rated Other proficiency accounts for unique variance (beyond self-rated English proficiency) in predicting English verbal fluency.  However, it remains to be seen if this generalizes to other measures of proficiency, fluency, vocabulary size, or vocabulary quality. 

Point 7:  Lines 49-57. This explanation of the rationale is a bit confusing. Can the authors clarify a bit? 

Response 7.  Yes.  Hope we did.

Point 8:  Line 107. “...and for their other language [age of acquisition] was 2.3 years”. I’m confused. Wasn’t this other language this subgroups’ native language? Why didn’t they start acquiring it until the age of 2.3 years? What did they do before that?

Response 8.  We hope that Responses 1 to 4 have cleared this up.  Other simply means the language of a bilingual that is not English as English was the test language for the verbal fluency task.   This other-than-English language will be a native language for some bilinguals, but a second language for others.   

Reviewer 3 Report

This study examines the role non-English language proficiency plays in explaining performance in an English verbal fluency task. In different models, non-English proficiency was indeed found to be related to the number of English responses bilinguals provided within a minute in a category/letter fluency task. The one exception was the model that also included scores from an objective measure of English proficiency.

The paper is well-written and the results are interesting. However, I have several main issues with the way the paper is presented and what it would actually contribute to our current literature and understanding of bilingualism and lexical access.

The paper presents two main theories that can explain bilingual disadvantage in verbal fluency: interference from the other language and 'weaker links' due to relatively less language use/exposure in each language. These are indeed the two main theories and it would be very interesting to see a study that tries to contrast the two theories. However, that is not what the current study does (or can do, given that it was not designed with this aim). The main finding can be interpreted by either theory. At the end, the authors use other studies on inhibitory control to suggest that the weaker links theory might be more applicable. Still, the current study itself does not contribute anything new to a better understanding of the mechanisms of verbal fluency disadvantages in bilinguals.

The interpretation of the first theory ('interference theory') seems to be that interference from the other language completely blocks access to specific words in the target language (which is given as a possible explanation as to why these words then cannot be used in the verbal fluency task). This is also used as the argument (page 8) why Green's ICM is not compelling with the interference account. However, I do not think that interference necessarily needs to work in the way of completely blocking specific lemmas. Instead, the hypothesis seems to be that interference between the languages slows down the production of a lemma, but does not hinder it entirely. If you only have one minute to name words and your production is slowed down by interference, a bilingual would indeed produce less words. Such a delay or slowing down due to interference is also more compatible with the typical finding that bilinguals are slower in e.g., picture naming. It is not the case that they cannot access the words at all, but rather that they are slower to do so.

Looking at the effects of proficiency on verbal fluency is presented as a novel question/method, but nothing is said about previous studies that have done the same. For example, Blumenfeld et al. (2016, International Journal of Speech-language pathology), Luo et al. (2010, Cognition), Friesen et al. (2015, Language, Cognition, and Neuroscience) are examples of other studies that have also assessed a link between verbal fluency and proficiency. These studies should at a very minimum be discussed, but the authors also need to explain how the current study is adding anything new to it.

The study talks about the D-KEFS switching measures, but then does not seem to do anything else with them. I would either leave this out or explain why it can be useful to interpret the data.

The data are collapsed across letter and category fluency based on the argument that in most previous studies they were not affected by bilingualism differently and based on the finding that average performance was the same in both tasks. It would be good to show too that in this dataset, category and letter fluency were affected similarly by bilingualism before collapsing across them.

Effects of 'other' (i.e., non-English) proficiency were not significant when English MINT scores were included in the model (i.e., an objective vocabulary measure rather than subjective self-ratings). The authors then show that MINT scores are partly explained by non-English proficiency, making the argument that including both MINT and non-English proficiency in the same model does not leave enough unique variance for non-English proficiency to explain. The authors then want to make the argument that this is not an issue when English self-rated proficiency scores are included in the model. To be honest, I do not follow this line of reasoning that is presented on page 6. Do the authors want to argue that only the English MINT is partly explained by non-English proficiency but self-ratings of English proficiency are not? In the same way that English MINT might be higher when non-English proficiency is lower, could you not expect English self-ratings to be higher when non-English proficiency is lower? At no point do I see any evidence in the form of actual data to support the authors' claim that English self-ratings are not influenced by non-English proficiency. At a minimum, some correlational analyses would be needed to support this claim.

Related to the previous point, page 6 talks about how bilinguals estimate their proficiency. The authors talk about different factors such as comparing yourself to bilinguals, to monolinguals, etc. A recent paper should be cited here that very clearly addresses this issue with actual data: Tomoschuk et al. (2019, Bilingualism: Language and Cognition). This study also shows the issue with using self-rated proficiency from a heterogenous group of bilinguals. This issue is very relevant here given that the bilingual group included many different bilinguals, speaking different languages and coming from different backgrounds.

Page 7 states that cognates do not compete. I would be careful with these claims given that while cognates may sometimes lead to facilitatory effects, there is also evidence for cognate interference and inhibition (e.g., Broersma et al., 2016, Frontiers).

The section on inhibition as a general construct is interesting, but given that the current data cannot do much to either support or criticise inhibition, this section is not contributing much. Furthermore, it again seems based on the assumption that interference has to work through inhibition (i.e., complete blocking of lemmas) rather than interference slowing down production (leading to fewer responses within a minute).

The section heading 'Next Steps and Limitations' is very confusing, because it does not say anything concrete about either next steps or limitations. 

Minor comments:

the last paragraph on page 1 introduces the two possible explanations for bilingual verbal fluency disadvantages. It would be good to see some references for each explanation. 

Page 2 presents the weaker links as a 'competing' hypothesis. I do not think that the interference and weaker links hypothesis are necessarily competing. An explanation in which both play a role is also quite plausible. 

Author Response

Response to Reviewer 3 Comments

Point 1:  Still, the current study itself does not contribute anything new to a better understanding of the mechanisms of verbal fluency disadvantages in bilinguals.

Response 1: We fully agree that our verbal fluency data cannot adjudicate between the interference and weaker links accounts of a bilingual disadvantage in verbal fluency, but would push back a bit on the contention that it adds nothing about cross-language effects in verbal fluency.  The original data set with 122 bilinguals and 108 monolinguals is much larger than typical past studies that often compared language groups with about 20 participants in each group.  This data, together with the systematic review of studies testing for a Language Group (monolingual vs bilingual) x Task (letter vs category) interaction provided reasonably compelling evidence that there is usually a bilingual disadvantage in verbal fluency and that its magnitude is usually the same across the two types of tasks.  These are, of course, the primary contributions of Paap et al. (2017, JCP) that provided the data for the regression analyses forming the core of this manuscript.  Sparked by the theme of this special issue we were able to tap into this unusually large database to verify that Other proficiency accounts for unique variance in English verbal fluency beyond that accounted for by English proficiency. To our knowledge, we are the first to formulate and test this hypothesis.  What follows from these findings is that there is a cross-language effect in verbal fluency.  Although interference and weaker links mechanisms have been proposed and debated as explanations for why bilingual disadvantages occur, another contribution of the present manuscript is developing an account of how weaker links could account for our regression findings. 

Point 2:  The interpretation of the first theory ('interference theory') seems to be that interference from the other language completely blocks access to specific words in the target language (which is given as a possible explanation as to why these words then cannot be used in the verbal fluency task). This is also used as the argument (page 8) why Green's ICM is not compelling with the interference account. However, I do not think that interference necessarily needs to work in the way of completely blocking specific lemmas. Instead, the hypothesis seems to be that interference between the languages slows down the production of a lemma, but does not hinder it entirely. If you only have one minute to name words and your production is slowed down by interference, a bilingual would indeed produce less words. Such a delay or slowing down due to interference is also more compatible with the typical finding that bilinguals are slower in e.g., picture naming. It is not the case that they cannot access the words at all, but rather that they are slower to do so.

Response 2:   This may be a case of strong intuitions running in opposite directions.  We have scored hundreds of tapes of verbal fluency and it is striking that participants who do relatively poorly often run out of responses fairly soon and sit in silence for the remainder of the one-minute interval – often breaking the silence to explain “I just can’t think of any more”.  Our intuition is that interference sometimes delays responses and sometimes blocks them.  For the categories we used (that produced means of about 11 correct responses) it appears that they are often blocked, at least in the sense of not being able to self-generate an effective retrieval cue.  One might also point out that latency is not always the primary dependent variable in picture naming; MINT, for example, assesses vocabulary knowledge as the number of correct responses. 

Point 3:  Looking at the effects of proficiency on verbal fluency is presented as a novel question/method, but nothing is said about previous studies that have done the same. For example, Blumenfeld et al. (2016, International Journal of Speech-language pathology), Luo et al. (2010, Cognition), Friesen et al. (2015, Language, Cognition, and Neuroscience) are examples of other studies that have also assessed a link between verbal fluency and proficiency. These studies should at a very minimum be discussed, but the authors also need to explain how the current study is adding anything new to it. 

Response 3.  Our study investigates the influence of the fluency in one language on verbal fluency in a second language.  The studies cited above do not.  Blumenfeld, Bobb, & Marian (2016) reported that English dominant bilinguals with higher Spanish proficiencies produced English cognates with lower frequencies compared to bilinguals with lower Spanish proficiency.  This may be an interesting result, but it is different from our research question.  Blumenfeld et al. do not report the simple correlation between Spanish proficiency and English verbal fluency, much less a regression that includes both L1 and L2 proficiency as predictors.  The primary purpose of Friesen, Luo, Luk, & Bialystok (2015) was to test for a language Group (bilingual vs monolingual) by fluency Task (letter vs category) interaction at various age levels.  No simple correlations or multiple regressions of L2 proficiency (Spanish for these bilinguals) on English (the dominant language for these bilinguals) verbal fluency were reported.  In the first paragraph of the introduction we discussed the systematic review and large-scale study presented in Paap et al. (2017) showing that a significant language Group x Task (letter vs category fluency), like that reported by Luo et al. is in a clear minority.  Luo, Luk, & Bialystok (2010) also focused on the Group x Task interaction, but in addition  reported that bilinguals with higher English vocabulary scores outperformed those with lower English vocabulary scores on English verbal fluency.  This is a moderately interesting finding, given that the self-ratings for the two groups were the same, but for present purposes showing that bilinguals with larger English vocabularies have better English verbal fluency than those with smaller English vocabularies is not relevant to the question of whether Other proficiency will affect English verbal fluency.  Luo et al. do not report correlations or regressions of L2 proficiency on L1 verbal fluencies.  Paap et al. (2017) discuss many of the other issues, especially the Group x Task interaction, at length and we could incorporate and extend that discussion to this manuscript, but our reaction is that it would require a lengthy digression away from the issues we would like to focus on.  In summary, our answer to what’s new in our study is that we are the first to show that Other proficiency accounts for unique variance in English verbal fluency beyond what English proficiency already accounts for. 

Point 4.  The study talks about the D-KEFS switching measures, but then does not seem to do anything else with them. I would either leave this out or explain why it can be useful to interpret the data.

Response 4.  The D-KEFS alternating targets task was important to the Paap et al. (2017) study  to determine if alternating verbal fluency would correlate with the three cued switching tasks in establishing a domain-general shifting ability.  (The switch costs for the three cued switching tasks were significantly correlated with one another, alternating categories correlated with alternating letters, but for the most part alternating fluency scores did not correlate with switch costs from the cued switching tasks.)  In any event, we agree that these conditions are not sufficiently relevant to present purposes and have deleted the mention. 

Point 5.  The data are collapsed across letter and category fluency based on the argument that in most previous studies they were not affected by bilingualism differently and based on the finding that average performance was the same in both tasks. It would be good to show too that in this dataset, category and letter fluency were affected similarly by bilingualism before collapsing across them.

Response 5.  The data from Paap et al. (2017) reanalyzed in this manuscript excludes the monolinguals.  However, Paap et al. (2017) did report that the Group (bilingual vs monolingual) x Task (category vs letter) was not significant in the full dataset. 

Point 6.  Effects of 'other' (i.e., non-English) proficiency were not significant when English MINT scores were included in the model (i.e., an objective vocabulary measure rather than subjective self-ratings). The authors then show that MINT scores are partly explained by non-English proficiency, making the argument that including both MINT and non-English proficiency in the same model does not leave enough unique variance for non-English proficiency to explain. The authors then want to make the argument that this is not an issue when English self-rated proficiency scores are included in the model. To be honest, I do not follow this line of reasoning that is presented on page 6. Do the authors want to argue that only the English MINT is partly explained by non-English proficiency but self-ratings of English proficiency are not?

Response 6.  As indicated in our response to Reviewer 1 we have revised this discussion and hope the logic is now clearer.  We are offering a conjecture that MINT scores are substantially influenced by Other proficiency, but that self-rated English proficiency scores are influenced by Other proficiency only when the frame of reference is monolinguals. 

Point 7.  In the same way that English MINT might be higher when non-English proficiency is lower, could you not expect English self-ratings to be higher when non-English proficiency is lower?

Response 7.  No.  Not if the self-ratings are based on comparisons between oneself and other bilinguals.  Based on the pattern of findings reported by Tomoschuk et al. we are speculating that most of our bilinguals use other bilinguals as a frame of reference.  (Please read the extended development in the revised manuscript.)

Point 8.  At no point do I see any evidence in the form of actual data to support the authors' claim that English self-ratings are not influenced by non-English proficiency. At a minimum, some correlational analyses would be needed to support this claim.

Response 8.  We are not claiming that English self-ratings are not influenced by non-English proficiency.  (If someone spends 80% of their time using Spanish, their actual English proficiency will suffer and lead to lower self-rated English proficiency.)  Rather, we are assuming that most of our bilinguals will use other bilinguals as their self-rating frame of reference and that any costs associated with cross-language effects will be canceled out because they appear in both parts of the comparison – that is, non-English proficiency has dampened performance of both the self-rater and the performance of the other bilingual comparison group.  As developed in the relevant section of the revised manuscript, these assumptions appear to be supported by the theory and some of the findings presented by Tomoschuk et al. (2019).  

Point 9.  Related to the previous point, page 6 talks about how bilinguals estimate their proficiency. The authors talk about different factors such as comparing yourself to bilinguals, to monolinguals, etc. A recent paper should be cited here that very clearly addresses this issue with actual data: Tomoschuk et al. (2019, Bilingualism: Language and Cognition). 

Response 9.  Many thanks for pointing out this highly relevant article by Tomoschuk et al. (2019).  As already stated, we not only cite it, but we think their theoretical approach to how people may generate self-ratings of proficiency is very compatible with our idea and that some of the findings (such as those involving the group of Chinese-English bilinguals who are recent immigrants to the USA) support or at least enhance the plausibility that many bilinguals will base their frame of reference on others like themselves rather than monolingual speakers. 

Point 10.  Page 7 states that cognates do not compete. I would be careful with these claims given that while cognates may sometimes lead to faciliatory effects, there is also evidence for cognate interference and inhibition (e.g., Broersma et al., 2016, Frontiers).

Response 10.  We have qualified our statement:  “cognates usually do not compete (but see Broersma, Carter, & Acheson, 2016, for exceptions).”

Point 11.  The section on inhibition as a general construct is interesting, but given that the current data cannot do much to either support or criticize inhibition, this section is not contributing much.

Response 11.  We agree that our regression analyses do not weigh in on the question of whether domain-general inhibition exists.  They do, however, show an adverse effect of Other proficiency on English verbal fluency.  Thus, the existence of domain general inhibition is extremely relevant to the debate as to whether bilingual disadvantages in verbal fluency are due to interference, weaker links, or a contribution of both.  We think that interference is the dominant view, because the questionable status of domain-free inhibition is underappreciated.  

Point 12.  The section heading 'Next Steps and Limitations' is very confusing, because it does not say anything concrete about either next steps or limitations. 

Response 12.  This section has been deleted.

Point 13.  The last paragraph on page 1 introduces the two possible explanations for bilingual verbal fluency disadvantages. It would be good to see some references for each explanation. 

Response 13.  Sandoval, Gollan, Ferreira, & Salmon (2010) are cited and that article discussed both explanations in great detail. 

Point 14.  Page 2 presents the weaker links as a 'competing' hypothesis. I do not think that the interference and weaker links hypothesis are necessarily competing. An explanation in which both play a role is also quite plausible. 

Response 14.  We completely agree and make the joint explanation explicit in several places.
